# One-Carbon Metabolism: Pulling the Strings behind Aging and Neurodegeneration

**DOI:** 10.3390/cells11020214

**Published:** 2022-01-09

**Authors:** Eirini Lionaki, Christina Ploumi, Nektarios Tavernarakis

**Affiliations:** 1Institute of Molecular Biology and Biotechnology, Foundation for Research and Technology-Hellas, 70013 Heraklion, Crete, Greece; lionaki@imbb.forth.gr (E.L.); ploumi@imbb.forth.gr (C.P.); 2Department of Basic Sciences, Faculty of Medicine, University of Crete, 70013 Heraklion, Crete, Greece

**Keywords:** aging, Alzheimer’s disease, diet, folate, metabolism, methionine, mitochondria, neurodegeneration, one-carbon vitamins, Parkinson disease

## Abstract

One-carbon metabolism (OCM) is a network of biochemical reactions delivering one-carbon units to various biosynthetic pathways. The folate cycle and methionine cycle are the two key modules of this network that regulate purine and thymidine synthesis, amino acid homeostasis, and epigenetic mechanisms. Intersection with the transsulfuration pathway supports glutathione production and regulation of the cellular redox state. Dietary intake of micronutrients, such as folates and amino acids, directly contributes to OCM, thereby adapting the cellular metabolic state to environmental inputs. The contribution of OCM to cellular proliferation during development and in adult proliferative tissues is well established. Nevertheless, accumulating evidence reveals the pivotal role of OCM in cellular homeostasis of non-proliferative tissues and in coordination of signaling cascades that regulate energy homeostasis and longevity. In this review, we summarize the current knowledge on OCM and related pathways and discuss how this metabolic network may impact longevity and neurodegeneration across species.

## 1. Introduction

One-carbon metabolism (OCM) refers to the complex network of biochemical reactions that mediate delivery of one-carbon units to various anabolic pathways [1]. The term “folate” describes a family of methyl donors that act mainly as enzymatic co-factors in vital interlinked anabolic pathways mainly required for DNA synthesis and amino-acid homeostasis. Folates share a common structure comprised of three chemically distinct moieties: the pteridine ring, the para-aminobenzoic acid (PABA) linker, and a tail of glutamic acid or polyglutamate. The pteridine ring and PABA are linked through a methylene bridge, which facilitates the binding of 1C units (attachment in the N-5 atom of the pteridine ring and the N-10 atom of the PABA linker). In contrast to plants, fungi, and certain prokaryotic organisms, which can synthesize de novo folate, mammals depend on its dietary uptake [2,3,4,5]. Uptake of folates across epithelia and into mammalian cells is facilitated by three transport systems, the Reduced Folate Carrier (RFC), the Proton-Coupled Folate Transporter (PCFT), and the Folate Receptors (FRs) [6,7]. PCFT is expressed mainly in the upper gastrointestinal tract and in tumors, supporting transport of folates under acidic conditions [8,9]. RFC is an organic anion antiporter expressed ubiquitously and is the major tissue folate transporter [10,11]. Folate receptors can be either the high affinity glycosylphosphatidylinositol (GPI)-anchored receptors (FRα and FRβ) that mediate folate transport via endocytosis or the soluble and secreted low affinity receptors (FRγ and FRδ) [12,13,14,15].

All natural folates differ in the oxidation state of the pteridine ring and are present in their reduced form, compared to the fully oxidized and monoglutamated folic acid, which is commonly used as a synthetic food supplement. Dietary absorption of folates occurs in the small intestine, where they are initially subjected to hydrolysis of their poly-glutamated tail to a mono-glutamated form. Once directed to the target cells, folate mono-glutamates are transformed to 5-methyl-tetrahydrofolate (5-methyl-THF), which is the predominant form of folates in animal tissues, and subsequently they are poly-glutamated again to participate in one-carbon reactions.

## 2. One-Carbon Metabolic Pathways

OCM is highly compartmentalized, since parallel, interconnected pathways exist in the cytosol, mitochondria, and nucleus [16]. In the following section we are analyzing in detail the OCM pathways in the distinct cellular compartments (summarized in Figure 1).

### 2.1. Cytosolic and Nuclear Pathways of One-Carbon Metabolism

Folic acid, the synthetic, fully oxidized, and monoglutamated form of folate, is first reduced to dihydrofolate (DHF) and then to tetrahydrofolate (THF) by DHF reductase (DHFR), which uses nicotinamide adenine dinucleotide phosphate (NADPH) as an electron donor. Cytosolic THF is then converted to 5,10-methylene-THF by the vitamin B6-dependent enzyme serine hydroxymethyltransferase 1 (SHMT1). This reaction is reversible and simultaneously converts serine to glycine [1].

Thymidylate synthase (TYMS) uses 5,10-methylene-THF to catalyze the methylation of deoxyuridylate (dUMP) to deoxythymidylate (dTMP) upon oxidation of a dihydroflavine-adenine dinucleotide (FADH_2_) molecule. TYMS-mediated de novo thymidylate generation is indispensable for DNA synthesis and is compartmentalized apart from the cytosol, in mitochondria and nucleus [17,18]. The removal of a 1C unit from 5,10-methylene-THF for dTMP synthesis converts the latter to DHF, which in turn can be transformed again to THF and 5,10-methylene-THF through the aforementioned reactions.

5,10-methylene-THF can be irreversibly converted to 5-methyl-THF by the vitamin-B2-dependent methylene-THF-reductase (MTHFR) in a process accompanied by oxidation of NADPH to NADP^+^ and the concomitant reduction of FAD to FADH_2_. Methionine synthase (MTR) in conjunction with vitamin B12 demethylates 5-methyl THF (conversion to THF) by donating its 1C unit to homocysteine (Hcy) and transforming it to methionine. MTR is regenerated to a functional state by Methionine Synthase Reductase (MTRR) after each cycle of methionine synthesis. The conversion of THF to 5,10-methylene-THF and then back to THF completes a series of reactions, known as the folate cycle. Methionine, produced by the remethylation of Hcy, is subsequently adenylated by methionine adenosyltransferase (MAT) to generate S-adenosylmethionine (SAM) by using an ATP molecule as a co-substrate. SAM is the major methyl donor for vital cellular processes and the master regulator of epigenetic mechanisms [19]. Donation of its methyl group converts SAM to S-adenosylhomocysteine (SAH), which in turn is converted to Hcy after removal of the adenosine molecule, through a reversible process mediated by S-adenosyl-L-homocysteine hydrolase (AHCY) [20]. In mammalian liver and kidney cells, Hcy can be recycled back to methionine through an alternative mechanism mediated by the Zn^2+^- and B6-dependent enzyme betaine-homocysteine methyltransferase (BHMT). This reaction uses betaine (trimethylglycine) as a methyl donor, which is acquired either by dietary uptake, or as a byproduct of choline degradation [21].

Sufficient levels of methionine enable the entry of Hcy in the transsulfuration pathway. B6-dependent Cystathionine beta synthase (CBS) catalyzes the first step by condensing Hcy and serine through a beta-replacing reaction, to finally generate cystathionine. Cystathionine is then converted to cysteine by cystathionine γ-lyase (CSE) [22]. Sequentially, cysteine is finally converted to glutathione through an energy-consuming, two-step enzymatic process. Firstly, glutamate cysteine ligase (GCL) couples glutamate to cysteine-generating G-glutamyl-cysteine. In turn, G-glutamyl-cysteine is conjugated to glycine through GSH synthase (GS), finally forming GSH [22]. The transsulfuration pathway is mainly regulated by SAM, which inhibits the MTHFR-mediated formation of 5-methyl THF, while it enhances CBS activity [23,24].

Apart from its involvement in the methionine/Hcy cycle, 5,10-methylene-THF can be utilized for formate and purine synthesis. In the cytosol, 5,10-methylene-THF, produced by SHMT1, is first converted to 5,10-methenyl-THF through the reduction of NADP+ to NADPH and then to 10-formyl-THF through hydrolysis. This is a reversible two-step process, catalyzed by methylene-THF dehydrogenase 1 (MTHFD1) through its methylene-THF dehydrogenase and methenyl-THF cyclohydrolase activities, respectively. Once formed, 10-formyl-THF can be reversibly converted to THF and formate, in an ATP-producing process mediated by the formyl C1-tetrahydrofolate synthetase activity of MTHFD1 [25,26]. Alternatively, 10-formyl-THF is utilized for de novo purine (adenine and guanine) synthesis. De novo purine synthesis is an 11-step process, resulting in the generation of inosine monophosphate (IMP) and its subsequent conversion to either guanosine monophosphate (GMP) or adenosine monophosphate (AMP). The main enzymes involved are glycinamide ribonucleotide transformylase (GART) and 5-aminoimidazole-4-carboxamide ribonucleotide formyltransferase/IMP cyclohydrolase (ATIC), and the process occurs in specialized multi-enzyme complexes, known as the purinosomes [27]. Additionally, 10-formyl-THF can be converted to THF in a process generating NADPH and CO_2_. This reaction requires the enzymatic activity of cytosolic aldehyde dehydrogenase 1 family member L1 (ALDH1L1) [28].

In the nucleus, folates are mainly used to support de novo dTMP synthesis for DNA replication. Similar to the cytosolic pathway, the activity of DHFR, TYMS, and SHMT1 is required for the interconversion between THF, 5,10-methylene-THF and DHF in the nucleus. SHMT2α, produced by alternative splicing of the serine hydroxymethyltransferase 2 (SHMT2) gene, functions redundantly to SHMT1 in the nucleus [29]. Interestingly, during the S and G_2_/M phases, the aforementioned enzymes are translocated to the nucleus, where they form a multi-enzyme complex in conjunction with proteins of the nuclear lamina, as well as with DNA replication and repair factors [30]. Cytosolic formate, mainly derived by the mitochondrial pathway, is also used as a nuclear one-carbon source for the production of 5,10-methylene-THF, requiring the activity of the MTHFD1 enzyme [31]. Post-translational modification with the small ubiquitin-like modifier (SUMO) protein is required for the nuclear translocation of TYMS, DHFR, SHMT1, SHMT2α, and MTHFD1 [30,32,33].

### 2.2. Mitochondrial One-Carbon Metabolism

Given the lack of DHFR activity in mitochondria, the mitochondrial folate cycle depends on the import of reduced monoglutamate folate forms, which could be either monoglutamate THF or 5-formyl-THF [34]. The import of monoglutamated folates across the inner mitochondrial membrane (IMM) is mediated by the specialized transporter MFT (mitochondrial folate transporter), encoded by the Solute Carrier Family 25 Member 32 (*SLC25A32*) gene [35]. Subsequently, monoglutamated folates are converted to polyglutamates by the mitochondrial isozyme of Folylpolyglutamate Synthase (FPGS), in order to support the mitochondrial folate cycle [36]. Similarly to the cytosolic pathway, serine is used as a one-carbon donor for the conversion of mitochondrial THF to 5,10-methylene-THF by SHMT2, the mitochondrial SHMT isozyme, which is also B6-dependent [35]. Mitochondrial transport of serine is mediated by the IMM protein sideroflexin 1 (SFXN1) [37]. In addition to serine, glycine, dimethylglycine (DMG), and sarcosine can be used as alternative one-carbon donors. Mitochondrial glycine is derived either by SHMT2-dependent reaction or by direct import from the cytosolic glycine pool by the specialized mitochondrial glycine carrier (GlyC), encoded by *SLC25A38* gene [38]. In turn, the mitochondria-localized glycine cleavage system (GCS) mediates the oxidative cleavage of glycine to CO_2_ and NH_3_, while the cleaved methylene group is accepted by THF and the latter is reversibly converted to 5,10-methylene-THF [39]. In mammalian liver cells, dimethylglycine and sarcosine, produced by choline oxidation, donate their methylene group to THF, through a process catalyzed by dimethylglycine dehydrogenase (DMGDH) and sarcosine dehydrogenase (SARDH), respectively [40]. Mitochondrial 5,10-methylene-THF can be utilized either for TYMS-dependent dTMP synthesis or for formate production through a series of reactions similar to the cytosolic pathway. Particularly, 5,10-methylene-THF is converted to 5-methenyl-THF and subsequently to 10-formyl-THF by methylene-THF dehydrogenase 2 (MTHFD2) or methylene-THF dehydrogenase 2 like (MTHFD2L) in a process necessitating reduction of NAD+ to NADH. MTHFD2L can use both NAD+ and NADP+ [41]. Recently, it was shown that MTHFD2 is required for de novo purine synthesis in mice and it is activated downstream of mTORC1 in an ATF4-depedent manner [42]. Moreover, serine acts as a source for NADH generation through the mitochondrial OCM. Specifically under conditions of impaired mitochondrial respiration, OCM-derived NADH is no longer consumed through the electron transport chain, thus leading to its aberrant accumulation and toxicity [43,44]. The 10-formyl-THF is subsequently converted to formate by methylene-THF dehydrogenase 1 like (MTHFD1L). Mitochondrial formate is then exported and incorporated to the cytoplasmic formate pool, which can be converted back to 10-formyl-THF and facilitate purine synthesis [26]. Importantly, due to its ability to interfere with mitochondrial respiration (inhibition of cytochrome c oxidase) [45], formate has to be rapidly effluxed from mitochondria, although the identification of a specialized formate exporter is still pending. Apart from formate synthesis, mitochondrial 10-formyl THF can be converted to THF by mitochondrial aldehyde dehydrogenase 1 family member L2 (ALDH1L2) [28] or it can be processed by methionyl-tRNA formyltransferase (MTFMT) to generate formylmethionyl-tRNAs (f-Met tRNAs), thus promoting mitochondrial protein synthesis [46]. A recent study in yeast cells showed that autophagy is required for preserving adequate levels of serine in mitochondria and for efficient Met-tRNA formylation [47]. Additionally, given that mitochondria lack MAT activity, mitochondrial import of cytosolic SAM is required to support mitochondrial methylation reactions. S-Adenosylmethionine Mitochondrial Carrier Protein (SAMC), encoded by the *SLC25A26* gene, mediates SAM transport into the mitochondrial matrix [48,49]. Mitochondrially imported SAM was recently shown to be specifically required for Complex I protein methylation and stability as well as iron sulfur cluster assembly [50].

Although cytosolic and mitochondrial branches are comprised of similar enzymatic reactions, which are often reversible, it is well accepted that they are uni-directionally regulated. Particularly, serine catabolism is favored in mitochondrial compartments, while serine synthesis is favored in the cytosol. The directionality of each pathway relies on the differential abundance of NAD(P)H and NAD(P)+ in the different compartments: Reduced NAD(P)H:NAD(P)+ ratio in mitochondria favors the activity of MTHFD2/ MTHFD2L enzymes towards 10-formyl THF formation and formate synthesis, while augmented NADPH:NADP+ ratio favors the activity of MTHFD1 towards formate catabolism and 5-10-methylene THF synthesis, which is required for the subsequent serine production [1].

The importance of the mitochondrial one-carbon pathway in preserving cytosolic folate is highlighted by a recent study conducted in mammalian cell lines. Ablation of mitochondrial folate metabolism by knocking down SHMT2 or MTHFD1L results in the elimination of cytosolic THF due to reversal of the cytosolic pathway towards formate formation [51]. In addition, products of mitochondrial metabolism, including ATP and other tricarboxylic acid (TCA) cycle metabolites, as well as products of mitochondrial OCM, including formate and serine-derived glycine, are required for cytosolic purine synthesis [52]. Notably, super-resolution microscopy has revealed that purinosomes are in close proximity to mitochondria, suggesting that they are actively positioned near centers of high ATP and metabolite abundance. Particularly, the same study provided evidence that mechanistic target of rapamycin (mTOR) governs the spatiotemporal control of purine synthesis [53].

## 3. One-Carbon Metabolism in Aging

OCM fine-tunes the abundance of critical metabolites in the cell, thereby regulating cell survival, growth, division, and stress response. It is known that metabolic dysregulation correlates and significantly contributes to several diseases, such as cardiovascular diseases, cancer, and inflammation, ultimately leading to increased frailty and death. Furthermore, signaling pathways such as caloric restriction (CR), reduced nutrient sensing, reduced insulin/insulin-like growth factor (IGF)-FOXO signaling, mTOR-AMP-activated protein kinase (AMPK) signaling, or proteostatic stress response pathways (endoplasmic reticulum (ER) unfolded protein response (UPR^ER^), mitochondrial UPR (UPR^mt^), integrated stress response (ISR), autophagy, DNA damage response (DDR)) often modulate metabolic procedures within cells and across tissues to improve health span and lifespan. Therefore, OCM which lies at the core of cellular metabolism, is expected to be a common denominator of these pathways, ultimately impacting aging and disease. In the following paragraphs we will discuss how several key metabolites of OCM or metabolic branches that directly interact with OCM pathways may associate with aging (summarized in Table 1).

### 3.1. Folate Cycle in Aging

Folate blood levels decrease in the elderly. However, little is known regarding the causal association between aging and folate deficiency. In mammalian animal models, folate metabolism has been reported to affect aging-related phenotypes. Comparison of aspects of folate metabolism between young and aged rats showed that aging affects folate levels in the serum (approximately 50% reduction) but does not have an effect in liver folate content [89]. Folate deprivation in diet leads to a decrease in folate levels both in the serum and the liver in both age groups. Moreover, folate deprivation is associated with increased poly-glutamate chain length in both young and old rats [89]. Folate-deprived mice exhibit circadian oscillation impairments similar to the ones exhibited by older individuals [90]. Mice fed on a folate-deprived diet from weaning until 8 and 10 months of age exhibit manifestations of brain aging such as short-term memory impairment, altered SAM metabolism, and acetylcholine levels. Moreover, these mice show differential expression of Presenillin-1, OCM genes, and epigenetic enzymes, possibly compensating for the reduced folate intake [91]. On the contrary, supplementation of several folate forms, in the context of a more complex dietary supplement, shows beneficial effects on lifespan in wt and transgenic growth hormone mice (TGM), which have been used as models for accelerated aging [81]. 

The most common polymorphism of the *MTHFR* gene in humans is the C677T (Ala222→Val), which produces an enzyme with reduced function. MTHFR mediates the conversion of 5,10-methylene-THF into 5-methyl-THF, and C677T polymorphism is accompanied with low folate and high Hcy levels. Interestingly, this polymorphism inversely correlates with several types of colorectal cancer and AD [92]. Moreover, C667T *MTHFR* displays a slightly beneficial effect on all-cause mortality in two large population studies [87,88].

Invertebrates are also folate auxotrophs. Integrated proteomic and metabolomic analyses in the eye of aged *Drosophila melanogaster* flies revealed that folate and purine metabolic pathways are affected the most, with strong downregulation of pteridine-containing metabolites [93]. The soil nematode *Caenorhabditis elegans* feeds on live *Escherichia coli* bacteria and depends on them for folate uptake. Pharmacological inhibition of folate synthesis in this nematode/bacteria system leads to increased nematode lifespan [62]. However, inhibition of *C. elegans* genes for folate uptake or polyglutamase activity does not exhibit significant benefits on nematode longevity [94]. Moreover, the antidiabetic drug metformin alters nematode lifespan by impacting bacterial folate and methionine metabolism. The effects of metformin on nematode longevity are reversed in worm mutants of *MAT* and *MTR* genes [69,95]. Glycine and serine supplementation also leads to extended lifespan in worms [65,68]. This extension is mediated by the methionine cycle as it is reversed in *MAT* (*sams-1*) and *MTR* (*metr-**1*) mutants [65]. Interestingly, transcriptional activation of glycine metabolism and folate-dependent OCM is a prominent trait of longevity shared between seemingly different longevity paradigms [65]. Recently, it was reported that folate intermediates undergo extensive quantitative changes that are often common among long-lived nematode mutants [63]. Folate, THF, and 5,10-methylene-THF are increased while 5-methyl-THF, 10-formyl-THF, and 5,10 methenyl-THF are robustly and reproducibly decreased. Similar data were obtained when the DHFR enzyme was genetically suppressed in nematodes, also leading to lifespan extension. Notably, supplementation with 5-methyl-THF reversed longevity of long-lived mutants, of reduced insulin signaling (rIIS), inherently caloric restricted, and mitochondrial mutants, while folic acid supplementation did not alter lifespan of wt nematodes [63]. Conclusively, increasing reports suggest that altered OCM may be a shared metabolic trait of longevity.

Unrepaired DNA damage that accumulates from development throughout adult-hood is correlated and may drive the aging process. Folate deficiency leads to imbalance in DNA precursors and misincorporation of uracil into DNA, causing DNA instability [96,97]. Moreover, gamma-irradiation, a well-established DNA damaging factor, leads to progressive depletion of folates due to their increased consumption for production of nucleotide bases [98] and the oxidative splitting of the folate molecule to pterin and p-aminobenzoylglutamic acid [99]. In mice, DHFR and TYMS activity peaks 92 and 76hours after whole body gamma irradiation, respectively, while MTHFR activity remains almost unchanged [100]. Folate deficiency in rats impairs DNA excision repair in colonic mucosa [101]. Additionally, aging affects folate levels and uracil misincorporation in the colon [102]. These findings pertinent to the interaction of folate metabolism with DNA synthesis and repair raise the possibility that OCM could affect the rate of aging by impacting genomic integrity. However, further research is required to elucidate the cause-and-effect relationship of folate metabolism, genomic integrity, and aging.

### 3.2. Methionine Cycle in Aging

CR without malnutrition is a universal strategy to increase lifespan across species [103]. A growing body of evidence supports that restriction of dietary proteins and amino acids, rather than carbohydrate and lipids, determines longevity upon CR [104]. Among them, methionine is an essential amino acid whose restriction from diet has been mostly successful in extending lifespan in yeast, invertebrate models, and rodents [55,75,76,83,84,85,105]. Genetic manipulations leading to methionine restriction yield similar results [56,57,58,69]. Interestingly, a comparative study of plasma methionine metabolic profile among 11 mammalian species whose lifespans range from 3.5 to 120 years revealed a correlation of longevity with reduced methionine and cystathionine levels [106]. Methionine restriction (MR) is proposed to reproduce some of the metabolic consequences of CR. For example, MR, like CR, leads to lower rates of mitochondrial reactive oxygen species (ROS) production and oxidative damage on mitochondrial DNA (mtDNA) in mice [107]. Suppression of the growth hormone (GH)/IGF somatotropic axis has been reported in rodents upon MR [83,104], leading to in vivo insulin sensitivity, reduced hepatic lipogenesis, and white adipose tissue lipid remodeling. Interestingly, the hepatic fibroblast growth factor FGF21, a component of the ISR pathway, mediates part of the effects [108,109,110]. Moreover, MR activates the phosphorylation of eukaryotic translation initiation factor 2A (EIF2A) in a general control nonderepressible 2 (GCN2) kinase-independent mechanism [111]. Reduction of SAM levels upon MR enables the SAM sensor upstream of mTORC1 (SAMTOR) to bind gap activity toward rags 1 (GATOR1) and thereby inhibit mTORC1 signaling [112]. Recently it was shown that dietary MR in mice leads to a rapid decrease in hepatic methionine, leading to inhibition of the translation initiation ternary complex, thereby slowing down protein translation. Moreover, MR in mice leads to ISR induction in a mTORC1- and activating transcription factor 4 (ATF4)-dependent manner [113]. A methionine-supplemented diet has been correlated with chromosomal and DNA damage in murine peripheral blood while MR rescues from basal DNA damage in murine liver [114]. Longevity in several organisms is mediated by nutritional, environmental, or genetic interventions that activate autophagy [115]. MR extends chronological lifespan (CLS) of yeast in an autophagy-dependent manner [56]. Interestingly, mitochondria-specific autophagic clearance, mitophagy, was reported to mediate the beneficial effects of MR on yeast CLS [54]. Finally, methionine supplementation can reverse the beneficial effects of glucose restriction on yeast replicative lifespan (RLS), suggesting that intracellular methionine may coordinate the outcome of several longevity-promoting pathways [116].

All these longevity-promoting signaling cascades regulate metabolic adaptation of cells and tissues in response to MR. However, apart from these pathways, methionine metabolism can affect longevity through impacting SAM availability. As mentioned above, 5-methyl-THF feeds into the methionine cycle to produce methionine and SAM. In *D. melanogaster,* SAM levels increase with age and enhancing its catabolism by glycine-*N*-methyltransferase (GNMT) overexpression, extends longevity [74]. Moreover, GNMT expression levels are enhanced in flies and mice upon reduced insulin/IGF signaling (rIIS), while GNMT is required for full rIIS longevity [117]. Reduction of SAM synthesis upon genetic inhibition of SAM synthase, the homolog of mammalian MAT, has also been proven beneficial for the lifespan of worms and flies [66,74]. Specifically in nematodes, suppression of *sams-1* or *sams-5* leads to longevity, possibly through distinct mechanisms [66,67]. Moreover, the SAM:SAH ratio is markedly increased, while liver and kidney *gnmt* mRNA levels are decreased in the short-lived growth hormone transgenic mice (GH-Tg). Methionine restriction reverses the SAM:SAH ratio in GH-Tgs but not to wt levels [118]. In flies, genetic suppression of two out of three AHCY homologs, systemically or in a tissue-specific manner, reduces age-related SAH accumulation and leads to lifespan extension [80].

The availability of SAM is a critical denominator of methylation reactions on DNA, RNA, and proteins, and represents a key conduit linking cellular metabolism to the epigenetic landscape. DNA methylation takes place in CpG dinucleotides, by the addition of a methyl group on the position 5′ of the cytosine pyrimidine ring. Methylation also occurs in the CpA, CpT, and CpC sites, although it is restricted to specific cell types such as neurons, glia, and pluripotent stem cells [119]. More rarely, it takes place in adenosine, producing *N*^6^-methyladenosine (m^6^A) [120]. DNA methylation is a fundamental signature of gene repression and has been used for the development of accurate age prediction tools, the so-called epigenetic clocks [121,122,123]. DNA methylation patterns are the cumulative result of de novo DNA methylation reactions, mediated by DNA methyltransferases 3A (DNMT3A) and B (DNMT3B) and maintenance of DNA methylation, mediated primarily by DNMT1 [120]. Interestingly, DNA methylation seems to decrease with age, with several human tissues showing similar age-related alterations [113]. Similarly, age-dependent hypomethylation is observed in rats and mice [124,125]. Moreover, age-associated methylation changes are suppressed in the long-lived Ames dwarf mice, which are deficient in the GH/IGF somatotropic axis [126]. DNA methylation changes relative to the exceptional longevity of bats occur primarily on innate immunity and tumorigenesis genes, suggesting that they exhibit enhanced immune response and cancer suppression [127]. Mitochondrial dysfunction, which escalates during aging, can also impact DNA methylation through methionine metabolism, as shown in a cell culture model of induced mtDNA depletion [128].

Histone methylation cooperates with DNA methylation to maintain chromatin epigenomic landscape. Imbalance of SAM:SAH ratio affects histone methylation, thereby modulating longevity in different species [105,129]. Suppression of transcription-activating histone marks (such as H3K4) or overexpression of suppressive histone marks (H3K27) are usually linked to longevity in invertebrates. Genetic perturbations of methyltransferases can impact longevity across species [105]. In mammals, further studies are required to decipher the role of histone methylation on lifespan. 

### 3.3. Homocysteine in Aging

Hcy is produced intracellularly in its reduced form and then it is exported by endothelial cells into the blood circulation where it is mostly oxidized to disulphides [130]. Elevation of Hcy plasma concentration represents a well-established risk factor for several age-related diseases such as cardiovascular dysfunction and stroke, while it also contributes to poor wound healing, loss of regenerative ability, and decline in renal and cognitive function [131]. Several factors can be linked to increased Hcy level. Low dietary folate and cobalamin (vitamin B12) intake leads to increased Hcy levels since they are required for methionine regeneration from Hcy. Renal dysfunction often arising with age affects proper removal of Hcy, leading to elevation of its serum levels [132]. Moreover, administration of certain anticonvulsant drugs or methotrexate, the specific inhibitor of DHFR, impairs folate metabolism, leading to the rise of serum Hcy [133,134]. As Hcy lies at the center of multiple metabolic reactions that regulate its conversions, dysfunction in the involved metabolic enzymes may impact its abundance. Therefore, dysfunction of MTR, MTRR, and MTHFR, which are required for Hcy remethylation to methionine, correlate with increased Hcy levels. Another pathway of Hcy remethylation to methionine is mediated by betaine and BHMT. Betaine is an amino acid derivative absorbed from diet and converted to trimethylglycine in enterocytes. Betaine is used as an osmolyte and a methyl donor for Hcy remethylation [131]. Interestingly, serum betaine is inversely correlated with weight, BMI, and body fat [135] while it linearly associates with a reduced risk of having low lean mass [136]. BHMT contains three cysteine residues that coordinate one Zn atom. Long-term exposure to oxidative conditions leads to cysteine oxidation, which hinders Zn binding and thereby enzyme activity in vitro [83].

### 3.4. Transsulfuration Pathway in Aging

When Hcy is not recycled to methionine it is channeled irreversibly through the transsulfuration pathway. In the canonical transsulfuration pathway, CBS mediates the condensation of Hcy and serine to form cystathionine and water. Depletion of the yeast CBS homolog *cys4* extends chronological lifespan (CLS) [59]. Ubiquitous or neuron-specific overexpression of CBS extends the lifespan of well-fed flies, while a functional CBS is required for lifespan extension upon dietary restriction [77,78]. In nematodes, robust proliferation of adult germline stem cells delays reproductive aging and leads to lifespan extension upon cold exposure. Interestingly, this longevity is mediated by prostaglandin E2 signals that induce cross-tissue activation of *cbs**-1* gene and hydrogen sulfide (H_2_S) production in the intestine [137]. Moreover, *cbs-1* overexpression in the nematode intestine or body wall muscles extends lifespan at standard (20 °C) and high (25 °C) rearing temperatures [137]. A recent study reported a unique plasma metabolic profile of centenarians characterized by increased transsulfuration pathway intermediates such as Hcy, cystathionine, and taurine [138]. When cysteine is used instead of serine, the products of CBS catalysis are cystathionine and H_2_S. There is increasing evidence for the role of H_2_S on aging regulation, although its effects on different species are contradictory [60,139,140]. Mammalian species with increased longevity have reduced H_2_S blood levels and liver CBS activity [141]. However, exogenous supplementation of H_2_S extends the lifespan of *C. elegans* by activating skinhead-1 (SKN-1) transcription factor [70,142] and in a CR-independent manner [67]. Recently it was shown that dietary restriction in mice leads to a CSE-dependent increase in H_2_S biosynthesis that leads to tissue-specific alterations in protein persulfidation [143]. Whether these changes in protein persulfidation are linked to longevity and stress resistance triggered by dietary restriction is a question that begs to be answered.

Another product of the transsulfuration pathway is taurine. Taurine is a conditionally essential beta-amino acid that is endogenously produced from cysteine, but mammals need to supplement its levels through dietary intake. It is one of the most abundant amino acids in the vertebrate brain in regions such as the cerebellum and the cortical cortex [144]. In humans, taurine supplementation associates with lower blood pressure, improved vascular function, reduced risk for ischemic heart disease, and other health benefits [145]. In mice, tissue-depletion of taurine in a taurine transporter knockout mutant leads to skeletal muscle deterioration and shorter lifespan in both male and female populations [86]. Taurine supplementation in nematodes leads to an extended lifespan [68], while in flies it rescues age-associated intestinal stem cell hyperproliferation and gut hyperplasia by eliminating UPR^ER^ stress [146].

### 3.5. Glutathione in Aging

GSH is the main intracellular redox buffer. It is composed of three amino acids, glycine, cysteine, and glutamic acid and it exists in a reduced and oxidized form. Cysteine produced from the transsulfuration pathway is usually the rate-limiting amino acid for GSH biosynthesis and GCL catalyzes the rate-limiting step in GSH biosynthesis. Once synthesized, GSH can be used as a co-substrate for glutathione peroxidase (GPX)-catalyzed reactions, mediating removal of hydrogen peroxide (H_2_O_2_) and lipid hydroperoxides and producing its oxidized form, the glutathione disulfide (GSSG). Reduction of GSSG back to GSH is catalyzed by glutathione reductase and uses NADPH [147]. The Redox Theory of aging proposed that longevity is modulated by the cellular redox state, which is reflected in the ratios of GSH/GSSG and NADP^+^/NADPH levels, among others [148]. Redox buffering systems do not only rescue macromolecules from oxidative damage but also regulate the levels of ROS molecules, which act as signaling components that modulate aging [149]. In yeast, depletion of *Gsh1* gene impacts CLS depending of the dietary glucose levels [61]. In *D. melanogaster*, systemic or neuronal overexpression of GCL catalytic or modulatory subunits increases glutathione content and extends their lifespan [79]. However, in worms, GSH supplementation has yielded contradicting results. Earlier studies have shown that small GSH concentrations prolongs longevity [72] while recently it was reported that prolonged supplementation of adult worms with dietary thiols such as N-Acetyl Cysteine (NAC) and GSH accelerated their aging [73]. Differences in the timepoint of NAC treatment could explain this discrepancy. Specifically, when NAC is provided from day three of adulthood onwards it is beneficial, while when it is provided just before (L4 larvae) and throughout adulthood it is not. This suggests that ROS could mediate a pro-longevity signal early in adulthood. If this signal is lost by early NAC supplementation, then NAC accelerates aging. If NAC is provided later in adulthood, it protects from age-related oxidative damage and thus it prolongs longevity. GSH blood levels drop in obese mice and humans; however, *Gclm^−^/^−^*
*mice*. Which lack the GCL enzyme, are protected against obesity and insulin resistance [150]. In humans, GSH supplementation has yielded poor benefits for the elderly, likely due to the instability of the molecule when transported through the gastrointestinal tract. Alternative approaches include supplementation with GSH precursor amino acids such as glycine and NAC or glycine and cysteine. This approach has yielded promising results in the potential treatment of age-related decline in the elderly [151,152,153].

## 4. One-Carbon Metabolism and Neurodegeneration

OCM, as a central metabolic hub for the production of critical metabolites that serve as building blocks for cellular growth and replication, it holds a very important role in regulating developmental processes and tumorigenesis [154,155,156,157]. Folic acid is typically prescribed to pregnant women to facilitate neuronal development of the fetus and prevent neural tube defects [158,159]. Nevertheless, its implication in preserving the integrity of post-mitotic neuronal cells is also under intense investigation. Many epidemiological studies associate reduced levels of folate with cognitive decline in the elderly [160,161,162]. However, whether these phenomena are causatively linked is not yet clear. Accumulating studies have associated OCM with the onset and development of neurodegenerative diseases. In the following section we are going to analyze how impairment in OCM contributes to Azheimer’s disease (AD) and Parkinson’s disease (PD), the most frequent types of age-related neurodegeneration.

### 4.1. One-Carbon Metabolism in the Pathogenesis of Alzheimer’s Disease

AD is one of the most prevalent types of dementia in the elderly worldwide. AD patients display progressive degeneration of neurons in the cerebral cortex and subcortical regions, including the hippocampus, a region associated with long-term memory storage. Extracellular deposition of aggregated amyloid-beta plaques, as well as intracellular formations of hyper-phosphorylated Tau proteins, known as neurofibrillary tangles, are suggested to be the main hallmarks for AD pathogenesis [163]. Several prospective studies during the last decades support that diet enriched in folates and vitamins is protective against AD pathogenesis [164,165,166,167,168,169].

Perturbations in OCM, as a consequence of inadequate dietary folate uptake, have been associated with initiation and progression of AD. Particularly, increased plasma Hcy and concomitant decrease of the SAM:SAH ratio in AD-affected brain samples imply disruption in methylation pathways, which lead to epigenome-dependent changes in the expression of disease-related factors [170,171,172]. Reducing folate and vitamin B12 uptake in neuroblastoma cell lines diminishes SAM levels and reduces methylation in promoter regions of beta-site amyloid precursor protein cleaving enzyme 1 (BACE1) and presenilin 1 (PSEN1) genes [173]. Both genes encode for proteins that participate in the amyloidogenic cleavage of amyloid protein precursor (APP) [174]. Demethylation enhances *BACE1* and *PSEN1* expression and promotes Aβ generation [173]. Hyperhomocysteinemia, imbalanced SAM:SAH ratio, and hypomethylation of *BACE1* and *PSEN1* genes were also confirmed in vivo, in mouse AD models upon deprivation of folate and vitamins B6 and B12. Remarkably, in the particular study, the affected mice displayed intra-neuronal Aβ accumulation and early manifestation of cognitive decline prior to extracellular Aβ deposition [175]. SAM administration in AD mouse models at this early stage restores global DNA and *BACE1* promoter methylation, reduces intra-neuronal and extracellular Aβ deposition, and reverses cognitive deficits in AD mouse models [176]. However, further studies are required for clarifying the association of hypomethylated *BACE1* regulatory regions and AD development, as results in post-mortem AD brain samples are contradicting [177,178]. On the contrary, despite the existence of limited contradicting studies regarding *PSEN1* gene, in vitro and in vivo studies have associated B-vitamin deficiency with enhanced *PSEN1* expression due to hypomethylation of its promoter [179,180]. Additionally, reduced CpG and non-CpG methylation patterns of *PSEN1* genomic regions have been reported in post-mortem human brains, although they do not correlate this phenotype with disturbances in OCM [181]. A point to be noted is that dietary supplementation with folic acid is also protective against Aβ generation by favoring α-secretase over β-secretase activity. Although mechanistic insights regarding DNA methylation are currently missing, it is reported that miR-126-3p, a miRNA that specifically targets the α-secretase A Disintegrin And Metalloproteinase Domain 9 (ADAM9), is significantly downregulated upon folic acid supplementation [182]. Deregulated DNA methylation status in AD-related contexts has also been attributed to differential expression and activity of specific methyltransferases and demethylases. Specifically, in vitamin B-deficient neuronal cells and AD mice models, expression and activity of Methyl-CpG Binding Domain Protein 2 (MBD2) demethylase are significantly increased. This phenotype is reversed upon SAM supplementation. Similarly, the activity of de novo methyltransferases DNMT3A and DNMT3B is increased in SAM-supplemented neuroblastoma cells and AD mice models [183]. It is noteworthy that folic acid administration in mice protects against Aβ accumulation by enhancing DNMT1 methyltransferase activity in a dose-dependent manner and reducing *PSEN1* and *APP* expression through methylation of their promoter regions [184], although the association of *APP* gene methylation with AD pathogenesis needs further validation, since several studies have reported non-significant changes [177,185]. Additionally, genetic polymorphisms in *MTRR* and *DNMT3A* genes have been associated with reduced D-loop methylation of mtDNA in AD contexts [186]. In accordance, the *MT-ND1* gene, which encodes for NADH-ubiquinone oxidoreductase chain 1, was found significantly hypomethylated in AD brain samples [187]. Generally, the findings regarding differential DNA methylation of AD-related genes should be interpreted with caution, as methylation status may be context dependent and vary during AD progression stages. 

Discrepancies in OCM have also been implicated in Tau phosphorylation and in the formation of pathogenic neurofibrillary tangles mainly through inhibition of Protein-phosphatase 2A (PP2A). PP2A is the main serine/threonine phosphatase involved in dephosphorylation of Tau protein in the brain and, thus, is considered to have a neuroprotective role against AD. Post-translational methylation of its catalytic subunit C positively regulates PP2A activity by stimulating its stability and substrate specificity [188]. Numerous studies have demonstrated reduced PP2A methylation in post-mortem AD brain tissues, in AD mouse models, as well as in neuronal cell lines [189,190,191,192,193,194]. Folic acid or SAM supplementation rescues PP2A hypomethylation and alleviates AD-related pathology [192,195,196]. A very recent study showed that methylated PP2A promotes the non-amyloidogenic cleavage of APP through interaction with non-receptor protein kinase Fyn [197]. Additionally, mice deficient in MTHFR, the enzyme that catalyzes the conversion of 5,10-methylene-THF to 5-methyl-THF, display decreased levels of methylated PP2A and concomitantly reduced expression of its methyl-transferase leucine carboxyl methyltransferase 1 (LCMT-1). The particular effects are aggravated upon folate deficiency [198]. An alternative mechanism for PP2A hypomethylation was suggested in a study conducted in rats upon intravenous administration of Hcy. High Hcy levels in the hippocampus enhances Tau phosphorylation by activating the PME methylesterase, which specifically demethylates PP2A and exacerbates Tau phosphorylation [199]. 

A growing amount of research has correlated disturbances in OCM and particularly hyperhomocysteinemia with induced oxidative stress in AD contexts [200,201,202,203,204,205]. Inadequate dietary uptake of folates and B vitamins [206], mutations in the gene of the rate-limiting enzyme MTHFR [207], as well as impairment in the transsulfuration pathway [208,209,210] are suggested to contribute to elevated Hcy levels in the blood of AD patients or in respective AD models. The major mechanism through which hyperhomocysteinemia induces oxidative stress and neurotoxicity is suggested to be the dysregulation of calcium homeostasis through overstimulation of NMDA receptors [200,211,212,213,214]. Hyperhomocysteinemia has also been linked to DNA damage-induced hypersensitivity to excitotoxicity and apoptosis in rat hippocampal neurons [215]. In accordance, patients deficient in CBS, a core enzyme for the transsulfuration of Hcy to cystathionine, display increased DNA damage due to Hcy accumulation [216]. Recent studies highlight the importance of adequate vitamin B12 intake in alleviating ROS-mediated oxidative damage and Aβ-induced paralysis in *C. elegans* AD models [217,218]. Importantly, they showed that this protective effect of B12 relies on its ability to promote Hcy remethylation by acting as a cofactor for MTR in the methionine cycle [218]. Interestingly, betaine supplementation in *C. elegans* is also protective against Aβ-induced toxicity, although this mechanism does not involve Hcy remethylation, but rather depends on the CBS-mediated transsulfuration pathway [219]. Conclusively, therapeutic strategies aiming to eliminate Hcy levels either through the methionine cycle or through the transsulfuration pathway have emerged to be beneficial for alleviating oxidative stress in AD. Deficiency in GSH, a well-established antioxidant produced by Hcy transsulfuration, is suggested to exacerbate AD-related oxidative stress and contribute to cognitive decline in AD patients [220,221,222,223].

### 4.2. One-Carbon Metabolism in the Pathogenesis of Parkinson’s Disease

PD is the second most prevalent age-related neurodegenerative disorder after AD. PD patients mainly display motor-related symptoms, including constant tremors, bradykinesia, impaired balance, and enhanced rigidity in their limbs and trunk. Motor dysfunction in PD develops by the progressive loss of dopamine-producing neurons of the substantia nigra (SN), a basal ganglia structure of the midbrain, predominantly associated with movement. Another major hallmark is the formation of intraneuronal protein clumps mainly comprised of alpha-synuclein (αSYN) and ubiquitin [224]. Although most PD cases are sporadic, mutations in several genes have been associated with PD onset. Among these, mutations in αSYN-encoding gene (*SNCA*), vacuolar protein sorting associated protein 35 (*VPS35*), and leucine-rich repeat kinase 2 (*LRRK2*) are the main causes of monogenic PD inherited in an autosomal dominant manner. There have also been identified autosomal-recessive cases of PD, mostly linked to mutations in *PARK2* (encoding for Parkin), phosphatase and tensin homolog-induced putative kinase 1 (*PINK1*) and *PARK7* (encoding for DJ-1) genes [225].

Numerous studies have demonstrated a potential role of OCM in the development of PD [226]. In accordance with AD, elevated plasma Hcy is evident in approximately 30% of PD patients and is suggested to be involved in the onset and progression of PD [227]. Hyper-homocysteinemia in PD patients may be aggravated due to the frequent administration of L-DOPA as a treatment to replenish dopamine loss. Particularly, L-DOPA acts as a methyl acceptor by using methyl groups from SAM, leading to increased SAH levels and subsequent elevation in Hcy [228]. An alternative mechanism for L-DOPA-mediated elevation in Hcy is the sequestration of vitamin B6, which leads to B6 deficiency and the subsequent inactivation of key enzymes participating in the transsulfuration pathway (CBS, CSE) [229,230]. Thus, it is suggested that the administration of folic acid or B vitamins may be beneficial for PD patients treated with L-DOPA [231]. Apart from the L-DOPA-dependent increase in Hcy, reduced dietary folate as well as impairment in the pathways that are involved in Hcy elimination (methionine cycle, transsulfuration pathway, Hcy release in the extracellular space) may also contribute to hyperhomocysteinemia in PD. Several case control and prospective studies show that folic acid and/ or vitamin B deficiency is evident in PD patients and is likely the major determinant of elevated Hcy in PD [232,233,234,235,236]. Interestingly, periodic administration of folic acid in PD patients normalized Hcy levels [237], while high dietary B6 intake was associated with a decreased risk for PD onset [234]. Although the exact roles of folate or B vitamins are still emerging, some previous studies display contradicting results and conclude that there is no association with PD onset and progression [233,238,239]. Furthermore, genetic polymorphisms in genes encoding the rate-limiting enzymes of the methionine cycle, MTHFR, MTR, and MTRR, have been associated with high risk for PD development and are suggested to contribute to hyperhomocysteinemia in PD patients [240,241,242].

Similarly to AD, several studies provide insights of dysregulated DNA methylation in PD [243,244], suggesting that disturbances in folate metabolism, the indispensable route for SAM production, may account for these epigenetic changes. Analyses of post-mortem PD brains or peripheral blood samples from PD patients show reduced DNA methylation in the promoter region or in intron 1 of *SNCA* gene, indicating that the particular hypomethylation contributes to elevated *SNCA* expression [245,246,247]. Although contradictory results support that hypomethylation of *SNCA* intron 1 does not interfere with the transcription of the *SNCA* gene [248], a subsequent study using advanced technical approaches confirmed that the particular methylation is cell-type specific and indeed it regulates *SNCA* expression [249]. Notably, reduced SAM levels are associated with elevated αSYN platelet formation in PD patients [250]. An interesting mechanism contributing to deregulation of DNA methylation in PD was suggested by a study conducted in post-mortem brain tissues, in neuronal cells as well as in αSYN-transgenic mice. They showed that nuclear levels of the maintenance methyltransferase DNMT1 are reduced due to αSYN-dependent sequestration of DNMT1 to the cytosol [251]. It would be interesting for future studies to investigate whether folate supplementation could rescue universal or *SNCA* gene DNA hypomethylation in PD contexts by reversing DNMT1 translocation or by enhancing the function of de novo DNMT3A and DNMT3B methyltransferases. However, DNA hypomethylation is not universal in PD contexts, since several genes, including PD-risk genes, are reported to be differentially methylated (and not only hypo-methylated as expected) [243]. These observations point out that there should be selective mechanisms determining hypo- or hypermethylation of specific promoters or genomic regions. In any case, perturbations in the SAM:SAH ratio evident in PD cases [250] surely influence the epigenetic outcomes. Notably, a very recent study showed that mtDNA is significantly hypomethylated in different tissues of PD patients compared to their control counterparts [252], in agreement with a previous report displaying reduced mtDNA D-loop methylation in the SN of PD patients [187].

A growing body of evidence has confirmed that mitochondrial dysfunction in conjunction with elevated oxidative stress both contribute to the degeneration of dopaminergic neurons in PD patients [253,254]. Importantly, a pronounced decrease in GSH levels is evident in the SN of PD patients [255]. Whether GSH deficiency is involved in PD aetiology or whether it is a consequence of it remains largely enigmatic. Additionally, it is not well defined whether disturbances in the transsulfuration pathway contribute to reduced GSH levels in PD contexts. Interestingly, inducible knockdown of *GCL* gene, encoding the rate-limiting enzyme for de novo GSH synthesis, results in selective inhibition of mitochondrial complex I and subsequent mitochondrial dysfunction in dopaminergic SN neurons of transgenic mice. This is mediated by an enhanced nitric oxide (NO)-dependent thiol oxidation event due to GSH loss [256]. Another study conducted in a *D. melanogaster* PD model showed that Parkin deficiency augments oxidative stress, curtails antioxidant activity, and disturbs mitochondrial function, as monitored by low ATP production. Notably, all these effects are ameliorated upon folic acid supplementation, supporting the beneficial role of folates in relieving PD-related oxidative stress and mitochondrial dysfunction [257]. Furthermore, shRNA against the PD-related *PARK7* gene in neuronal cells leads to elevation of oxidative stress and significant reduction in *SHMT2* and *MTHFD2* genes [258]. Both genes are implicated in mitochondrial serine catabolism and formate synthesis, suggesting that formate deficiency could contribute to *PARK7*-related PD pathogenesis. 

## 5. Conclusions and Future Perspectives

Our knowledge on metabolic adaptations that take place upon aging and neurodegeneration has greatly advanced during the last years, due to the development of new analytical methods. We now appreciate that specific metabolites and metabolic pathways are typically associated with old age and/or disease. OCM is a central metabolic hub and holds a prominent role in the regulation of cellular metabolic status. However, the more we deepen into the complexity of the metabolome, we appreciate the need to further elucidate the relative metabolic fluxes that characterize longevity, aging, and disease. On top of this complexity, tissue-specific and disease-specific contexts should be considered. As these fluxes emerge, new therapeutic strategies will develop to control them in specific tissues and rewire the metabolic profile to the benefit of the tissue and the organism. OCM is a promising target of such therapeutic interventions as it directly responds to dietary inputs and pharmacological compounds. Future studies should determine the exact mechanisms through which OCM could safely combat age-related decline.

## Figures and Tables

**Figure 1 cells-11-00214-f001:**
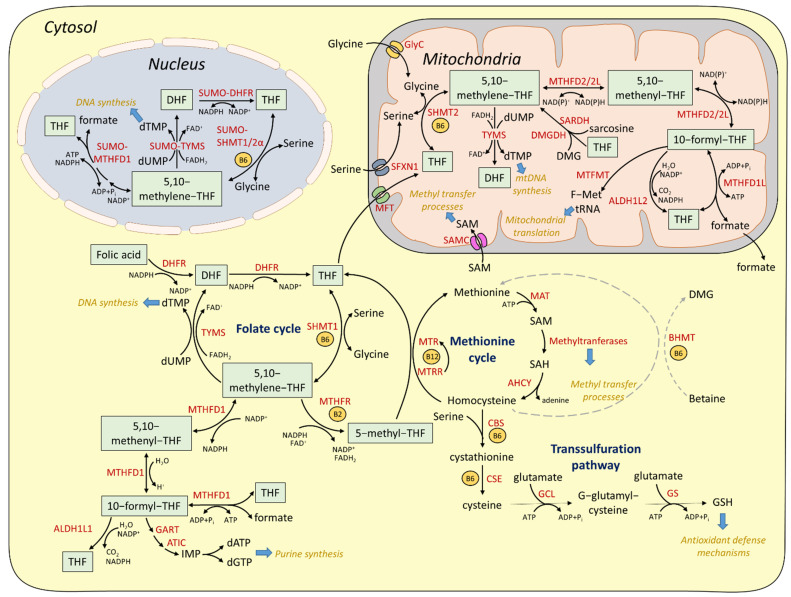
One−carbon metabolic pathways. Intersections of OCM pathways in cytosolic, nuclear, and mitochondrial compartments, based on mammalian systems. The involved folate forms are depicted in green boxes, while the respective enzymes are written in red.

**Table 1 cells-11-00214-t001:** Genetic and environmental manipulations that impact lifespan via modulation of one-carbon metabolism and associated pathways. CLS, chronological lifespan; RLS, replicative lifespan; NAC, N-Acetyl Cysteine; CR, caloric restriction; GSH, glutathione; MTHF5, 5-methyl-tetrahydrofolate.

Genetic Manipulation	Environmental Manipulation	Effect on Lifespan	References
**Yeast**			
	Methionine restriction	CLS extension	[54,55]
*met2* deletion		CLS extension	[56,57]
*met15* deletion		CLS extension	[56,57]
*met3* deletion		RLS extension	[58]
*sam1* deletion		RLS extension	[58]
*cys4* deletion		CLS extension	[59]
	H_2_S	CLS extension	[60]
*gsh1* deletion	10% glucose	CLS shortening	[61]
**Nematodes**			
	sulfamethoxazole	Extension	[62]
*dhfr-1* RNAi		extension	[63]
*tyms-1* RNAi		extension	[63,64]
*daf-2(e1370)*	MTHF5 supplementation	Reverses longevity *of daf-2* mutants	[63]
*mel-32* RNAi		extension	[65]
*sams-1* RNAi		extension	[66,67]
*sams-5* RNAi		extension	[67]
	Glycine supplementation	extension	[65,68]
	Serine supplementation	extension	[65,68]
	Metformin	extension	[69]
*sams-1* mutant	Metformin	Reverse of metformin’s benefits	[69]
*metr-1* mutant	Metformin	Reverse of metformin’s benefits	[69]
	H_2_S	extension	[70,71]
*cbs-1* overexpression		extension	[60]
	NAC from day 3	extension	[72]
	NAC from L4	shortening	[73]
	taurine	extension	[68]
	Acivicin (GSH restriction)	extension	[73]
**Flies**			
*GNMT* overexpression		extension	[74]
	Methionine restriction	extension	[75,76]
*Sams* depletion		extension	[74]
*Cbs* overexpression		extension	[77]
*Cbs* depletion	Caloric restriction	Reversed CR-driven longevity	[78]
*Gclc* overexpression		extension	[79]
*Gclm* overexpression		extension	[79]
*dAhcyL1/dAhcyL2* suppression		Extension	[80]
**Mammals**			
Transgenic growth hormone mice	Supplementation of several forms of folate	Extension	[81]
wt	Supplementation of several forms of folate	Extension	[81]
Female SHR mice	metformin	Extension	[82]
Rats, mice	Dietary Methionine restriction	Extension	[83,84,85]
Tissue-specific taurine transporter depleted mice		Shortening	[86]
**Humans**			
C667T *MTHFR*		Associates with decrease in all-cause mortality	[87,88]

## Data Availability

Not applicable.

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
