# Peer review of "One-Carbon Metabolism: Pulling the Strings behind Aging and Neurodegeneration"

_cells, 2022, doi:10.3390/cells11020214_

Round 1
Reviewer 1 Report
the review proposed is a well written overview on One Carbon Metabolism (OCM) paradigms witha a focus on folate and methionine cycle. The manuscript is properly organized and causes and implications of OCM dysregulation in ageing and neurological disease are appropriately described. However, I would recommend to include a brief overview about the impact of genetic/genomic alterations in the overall OCM activation and/or regulation.
Author Response
We thank the reviewer for their positive evaluation on our manuscript and for the suggestion to include genetic/genomic alterations pertinent to OCM activation/regulation. The effect of OCM to genomic stability through impacting DNA precursor biosynthesis, DNA repair and epigenetic alterations has been extensively studied in the context of development and tumorigenesis and previously reviewed. Although, it is not surprising that OCM could also affect the ageing process through impacting genomic stability, to our knowledge their direct association with the ageing process has not been directly assessed. We have added more information on genetic/genomic alterations pertinent to OCM in lines 274-287, 313-315 and 468-469.
Reviewer 2 Report
This review summarizes the recent findings on OCM and its effects on ageing and neurodegeneration.
The authors gave a very comprehensive description of one-carbon metabolic pathways, including the specific aspects in different compartments, eg. the similarities and the differences of OCM in mitochondria vs. in the cytosol. Then they discussed about the effects of different metabolic pathways in ageing and in neurodegenerative diseases such as Alzheimer's and Parkinson’s diseases, and listed recent findings in model organisms and in humans. This review was well written and organized.
The discussion was thorough and compiled the most significant findings in one synthesized review. It would be a nice addition to the literature of this field.
Author Response
We appreciate the nice words of the Reviewer for our manuscript.
Reviewer 3 Report
Although this not the first review on OCM and neurodegeneration, the manuscript by Lionaki and colleagues appears really well organized, clear and comprehensive. It is anbd easy lecture reporting, at the same time, key and specific information.
I just have few minor comments:
1) the CLS/RLS acronyms are explained only at page 8 but theu are firstly reported in table I at page 5. The abbreviations should then be made explicit at the bottom of the table.
2) describing the non-CpG methylation as a "rare" modification (lines 314-316) is no longer acceptable after many papers evidenced large and functional non-CpG methylation in adult and somatic cells, particularly in the nervous tissue. I suggest to review this statement and briefly discuss the non-CpG methylation in aging and neurodegeneration.
3) The recap on PSEN1, BACE1 and APP methylation/expression in mice or cells under OCM modulations deserves some more clarity and some correction:
a) BACE1 promoter differential methylation deserves further studies since it has been observed in one of the cited studies but not in others
b) on the contrary, methylation-dependent PSEN1 modulation has been further associated to the AD-like phenotype in mice under OCM alterations and alsdo observed in human brain tissue (Fuso et al, NBA 2012; Monti et al., Epigenetics 2020)
c) APP differential methylation has weak evidence and rather was not replied in several studies
Author Response
Although this not the first review on OCM and neurodegeneration, the manuscript by Lionaki and colleagues appears really well organized, clear and comprehensive. It is anbd easy lecture reporting, at the same time, key and specific information.
We thank the Reviewer for their positive comments on our manuscript.
I just have few minor comments:
1) The CLS/RLS acronyms are explained only at page 8 but theu are firstly reported in table I at page 5. The abbreviations should then be made explicit at the bottom of the table.
We have included the full name of the respective acronyms in Table 1, as suggested by the Reviewer.
2) Describing the non-CpG methylation as a "rare" modification (lines 314-316) is no longer acceptable after many papers evidenced large and functional non-CpG methylation in adult and somatic cells, particularly in the nervous tissue. I suggest to review this statement and briefly discuss the non-CpG methylation in aging and neurodegeneration.
We thank the Reviewer for their point on CpG methylation. We have incorporated the respective changes in lines 341-344 and 558-561.
3) The recap on PSEN1, BACE1 and APP methylation/expression in mice or cells under OCM modulations deserves some more clarity and some correction:
- a) BACE1 promoter differential methylation deserves further studies since it has been observed in one of the cited studies but not in others
We have made the suggested corrections and have added the contradicting studies regarding BACE1 promoter methylation (lines 499-503).
- b) on the contrary, methylation-dependent PSEN1 modulation has been further associated to the AD-like phenotype in mice under OCM alterations and alsdo observed in human brain tissue (Fuso et al, NBA 2012; Monti et al., Epigenetics 2020)
We have emphasized that methylation-dependent PSEN1 modulation has been associated with AD-pathogenesis by adding the suggested references (lines: 503-508).
- c) APP differential methylation has weak evidence and rather was not replied in several studies
We have mentioned in the text that APP differential methylation needs further investigation (lines: 523-524).